# Molecular Regulation of Autophagy and Asymmetric Cell Division by Cancer Stem Cell Marker CD133

**DOI:** 10.3390/cells12050819

**Published:** 2023-03-06

**Authors:** Hideki Izumi, Yasuhiko Kaneko, Akira Nakagawara

**Affiliations:** 1Laboratory of Molecular Medicine, Medical Research Institute, Saga Medical Center KOSEIKAN, Saga 840-8571, Japan; 2Research Institute for Clinical Oncology, Saitama Cancer Center, Saitama 362-0806, Japan; 3Saga HIMAT, Tosu 841-0071, Japan

**Keywords:** CD133, asymmetric cell division, autophagy, β-catenin, neuroblastoma

## Abstract

CD133, also called prominin-1, is widely known as a cancer stem cell marker, and its high expression correlates with a poor prognosis in many cancers. CD133 was originally discovered as a plasma membranous protein in stem/progenitor cells. It is now known that Src family kinases phosphorylate the C-terminal of CD133. However, when Src kinase activity is low, CD133 is not phosphorylated by Src and is preferentially downregulated into cells through endocytosis. Endosomal CD133 then associates with HDAC6, thereby recruiting it to the centrosome via dynein motors. Thus, CD133 protein is now known to localize to the centrosome as endosomes as well as to the plasma membrane. More recently, a mechanism to explain the involvement of CD133 endosomes in asymmetric cell division was reported. Here, we would like to introduce the relationship between autophagy regulation and asymmetric cell division mediated by CD133 endosomes.

## 1. Introduction

Cancer cells comprise heterogeneous cell populations in which genetic mutations accumulate, leading to more malignant cell populations. However, cancer stem cells with a high malignant potential have been identified in cancer cell populations, suggesting that such cells may be the key to generating tumor cell heterogeneity and resistance to drugs and radiation [1]. Although cancer stem cells have been shown to express specific cell surface markers, functional analysis of these markers has been limited due to the very low proportion of cancer stem cells in tumor cell populations [1].

CD133 is a pentaspan transmembranous protein that primarily localizes to the plasma membrane of normal and cancer stem cells and is widely known as a cancer stem cell marker [2,3]. In 2019, we reported that CD133 is downregulated into cells by endocytosis, transported to the centrosome, and inhibits autophagy [4], and that the centrosome-localized CD133 maintains cancer cells in an undifferentiated (cancer stem-like) state [4]. Recently, we also found that the distribution of CD133 endosomes, which localize to the centrosome, is highly heterogeneous among cells, and that cells exhibiting various autophagic activities are generated by asymmetric cell division [5]. In addition, it was found that β-catenin is nuclear-localized in daughter cells with a higher CD133 endosome distribution than daughter cells with a low CD133 endosome distribution. Thus, β-catenin contributes to the malignant phenotype of cells in addition to the function of CD133 [5].

Here, we introduce recent progress in autophagy regulation and asymmetric cell division mediated by CD133 endosomes.

## 2. Cancer Stem Cell Marker CD133

The cancer stem cell marker CD133 was initially identified as a cell surface marker of human hematopoietic stem cells and mouse neuroepithelial cells [6,7,8]. It was subsequently reported to function as a marker of cancer stem cells in solid tumors, such as colon cancer, glioblastoma, hepatocellular carcinoma, and neuroblastoma [2,3]. CD133-positive cell populations have better self-renewal capacity and chemotherapy resistance characteristics than CD133-negative cell populations. In addition, CD133 expression was reported to be associated with the tumor grade and a poor prognosis [2,3].

CD133 is phosphorylated in its intracellular C-terminal region by Src family tyrosine kinases [9], resulting in the binding to and activation of the p85 subunit of phosphoinositide 3-kinase (PI-3K), which then signals downstream targets such as Akt in cancer stem cells [10] (Figure 1A). CD133 is also stabilized by binding to histone deacetylase 6 (HDAC6), which increases the transcriptional activity of β-catenin, and promotes cell proliferation and inhibits cell differentiation [11] (Figure 1A).

## 3. CD133 Localizes to Centrosomes and Suppresses Autophagy Activation

To elucidate a novel mechanism involving CD133, we searched for cancer cell lines that express high levels of CD133. As a result, we identified Huh-7, a hepatocellular carcinoma cell line, and SK-N-DZ, a neuroblastoma cell line, as cancer stem cell models. We found that CD133 is mainly localized to the centrosome, rather than the plasma membrane, in Huh-7 and SK-N-DZ cells [4]. Further analysis showed that when Src family kinase activity is low, CD133 is not phosphorylated by Src, resulting in the intracellular downregulation of unphosphorylated CD133 by endocytosis. The unphosphorylated endosomal CD133 then preferentially interacts with HDAC6, and CD133 endosomes are transported to the centrosome via the dynein motor [4] (Figure 1B). The centrosome is the major microtubule organizing center (MTOC) of animal cells and plays important roles in regulating cell polarity and motility, spindle formation, chromosome segregation, and cell division [13]. Recently, centrosomes were also reported to be involved in the regulation of autophagy [14].

Autophagy is a highly conserved protein/organelle degradation system. It is responsible for the turnover of long-lived proteins and the disposal of excess or damaged organelles in order to maintain cell homeostasis [15,16]. ULK1 is at the top of the autophagy cascade, and GABARAP (GABA receptor-associated protein) is necessary for the activation of ULK1-mediated autophagy initiation. The activated-ULK1 then activates the autophagy initiation complex, and elongation of the isolation membrane also occurs [16,17]. The isolation membrane then closes and engulfs cytoplasmic constituents, forming an autophagosome. Finally, the autophagosome fuses with a lysosome, resulting in complete degradation by lysosomal enzymes [15,18].

We subsequently investigated the biochemical functions of centrosome-localized CD133, and found that it traps GABARAP, an important regulator of autophagy initiation, to inhibit GABARAP-mediated ULK1 activation, and the subsequent initiation of autophagy (Figure 1B) [4]. The amino acid sequence of CD133 (828–831 aa (YDDV)), which contains a phosphorylation site (Y828) by Src kinase, is conserved as an LC3B-interacting region (LIR: Y/FXXV) [19], which is also a GABARAP-interacting motif. In fact, the phosphorylation form of CD133 reduces the ability to bind to GABARAP [4]. Therefore, it is likely that CD133 captures (traps) GABARAP via this potential LIR at centrosomes.

In addition, the physiological functions of centrosome-localized CD133 were investigated: CD133 suppressed autophagy and inhibited cell differentiation, such as primary cilia formation and neurite outgrowth [4]. Since autophagic activity is essential for the differentiation of stem cells, such as neural and embryonic stem cells [20,21], it was suggested that CD133, which localizes to centrosomes, may maintain cancer (stem) cells in an undifferentiated state by inhibiting autophagy [4].

## 4. CD133 Heterogeneously Localizes to Centrosomes and Generates Asymmetric Cell Division, Resulting in Daughter Cells Exhibiting a Variety of Autophagic Activities

We further examined the subcellular localization of CD133 in detail using SK-N-DZ cells, and found that the distribution of CD133 centrosome localization was heterogeneous in different cells [5]. Considering our above-mentioned finding that the centrosome localization of CD133 suppresses autophagy, the heterogeneous distribution of CD133 indicates that cells exhibiting a variety of autophagic activities are distributed. Indeed, knockdown of CD133 in cells with centrosome-localized CD133 increases autophagic flux, while forced expression of CD133 causes the centrosome distribution of CD133 and decreases autophagic flux [5]. Subsequent analysis revealed that the distribution of CD133 is inversely correlated with that of p62 (SQSTM1), a receptor for selective autophagy, and that CD133 binds to p62 and is degraded by selective autophagy [5].

Next, we investigated the mechanism of the formation of CD133 heterogeneity, and found that CD133 is apart from the centrosome when cells enter mitosis (prophase), and is then re-localized to the centrosome at the end of mitosis (cytokinesis), where CD133 endosomes may be distributed asymmetrically to create CD133 heterogeneity (Figure 2) [5]. Asymmetric cell division is a characteristic of cancer stem cells as well as normal stem cells, and is a very ingenious strategy because it maintains appropriate numbers of self-renewal stem cells and differentiated cells in a single division.

During CD133-mediated asymmetric cell division, β-catenin becomes nuclear-localized in daughter cells with a high CD133 centrosome distribution, while in daughter cells with a low CD133 centrosome distribution, β-catenin remains at the plasma membrane (Figure 2A) [5]. Furthermore, the expression of cyclin D1, a transcriptional target of nuclear-localized β-catenin, showed asymmetry similar to that of β-catenin [5]. β-Catenin has been reported to suppress p62 gene expression via TCF4, while it is degraded through selective autophagy via the formation of LC3 complexes [22]; these results suggest that CD133 and β-catenin cooperate to inhibit the autophagy-related p62 function and generate cell asymmetry (Figure 2B).

## 5. Conclusions and Perspective

In this commentary, we have outlined advancements in knowledge of the emerging functions of CD133, focusing on our studies. The importance of centrosome-localized CD133 has been suggested by the fact that it is observed in a large number of clinical specimens [4]. Plasma membrane-localized CD133 is mainly associated with PI-3K-mediated signal transduction for cell proliferation, whereas centrosome-localized CD133 suppresses autophagic activity and enhances the undifferentiated (stem-like) state of cancer cells, suggesting that CD133 is a multifunctional protein that exerts its function according to its subcellular localization.

More recently, we showed that centrosome-localized CD133 induces asymmetric cell division, resulting in autophagy-based tumor cell heterogeneity. What is the biological importance of producing daughter cells with different autophagic activity? Our findings suggest that autophagy-based asymmetric cell division plays an important role in tumor cell heterogeneity. The production of cells with different autophagic activity may lead to the constant production of cancer cells that can adapt to various extracellular environments.

Why do CD133 endosomes become asymmetrically distributed during cell division? Generally, recycling endosomes are present in the midzone in the telophase and are then transported to the centrosome by cytokinesis [23]. The mechanism is not yet known, but it is likely that CD133 endosomes are symmetrically distributed in most cases, but due to some stochastic timing, approximately 20% of cells may show an asymmetric distribution. Alternatively, inheritance of the mother or daughter centrosome might be a key process to determine symmetry-breaking in cell division. The inheritance pattern of mother/daughter centrosomes during asymmetric cell division is an important issue [24,25], and further study is necessary for its complete elucidation.

CD133 was originally known as a plasma membrane protein and may be a receptor for some ligands, and there are still unknown aspects of its function. As functional analysis of CD133 progresses, unexpected functions may be revealed. We look forward to further progress in this area of research.

## Figures and Tables

**Figure 1 cells-12-00819-f001:**
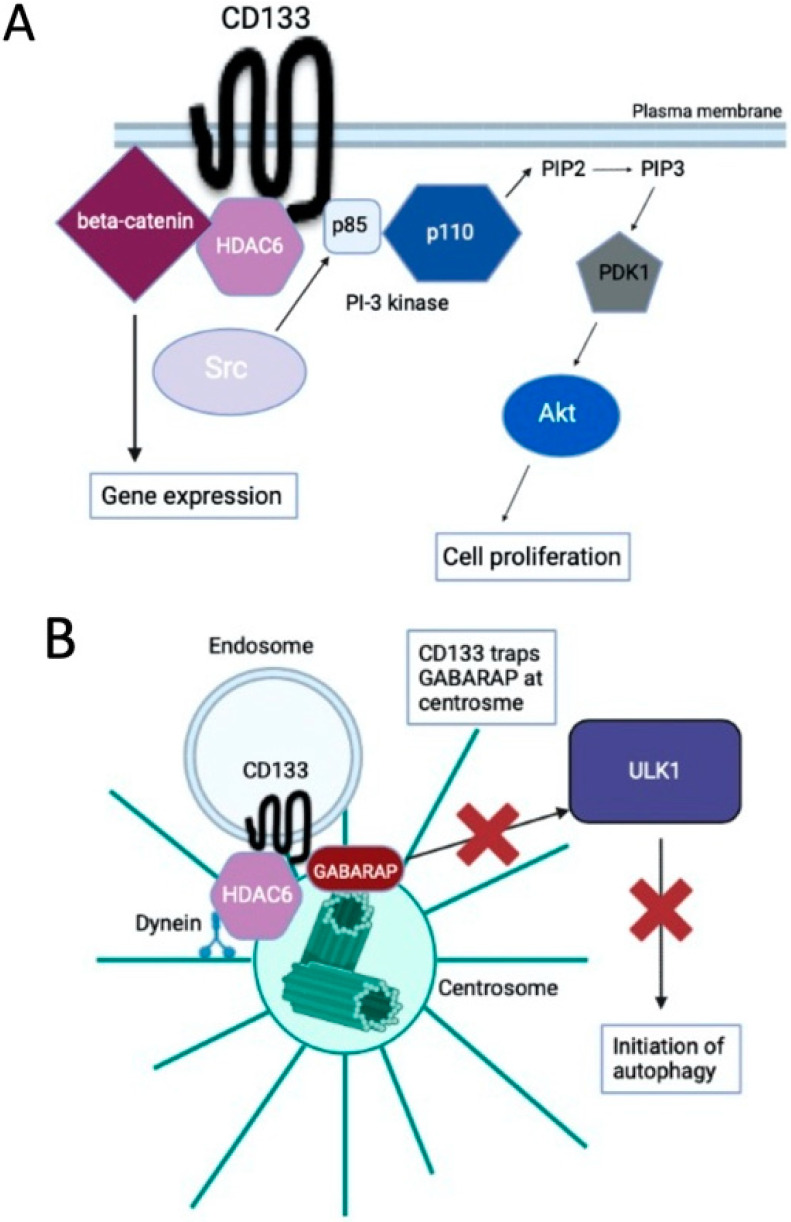
The signal transduction pathway of CD133. (**A**) CD133 is phosphorylated in its intracellular C-terminal domain by Src family tyrosine kinases, resulting in the binding and activation of the p85 subunit of phosphoinositide 3-kinase (PI-3K). Activated PI-3K then signals downstream targets such as Akt to promote cell proliferation. CD133 is also stabilized by binding to histone deacetylase 6 (HDAC6), which increases the transcriptional activity of β-catenin, promotes cell proliferation, and inhibits cell differentiation. (**B**) Under conditions in which Src family kinases are inactivated, unphosphorylated CD133 is transported from the plasma membrane to intracellular regions via endocytosis. After endocytosis, CD133 endosomes are transported along microtubules to centrosomes, assisted by HDAC6 and dynein motors. Endosomal CD133 localizes to the centrosome, which traps GABARAP, and inhibits the GABARAP-ULK1 interaction and initiation of autophagy. These figures were partly modified from the *Journal of Japanese Biochemical Society* **2022**, *94*, 415–418. doi:10.14952/SEIKAGAKU.2022.940415 [12].

**Figure 2 cells-12-00819-f002:**
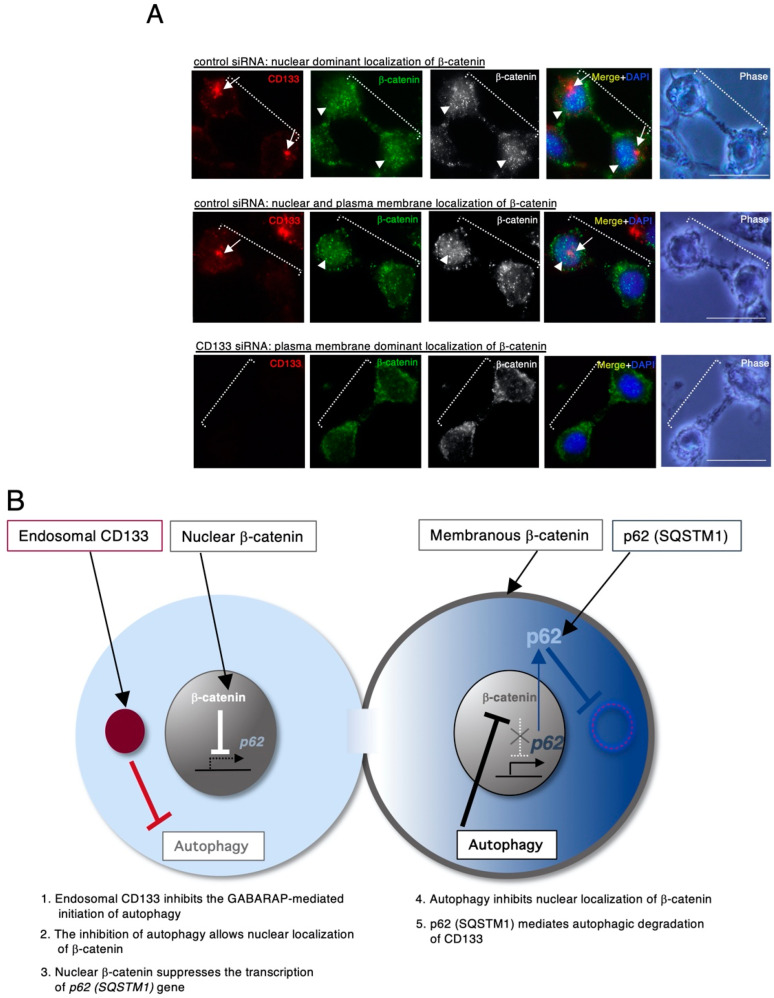
Symmetry-breaking of autophagic activity during cytokinesis in CD133-positive human neuroblastoma cells. (**A**) Representative images of CD133 and β-catenin during cytokinesis in SK-N-DZ cells transfected with control or CD133 siRNA. CD133 is red, β-catenin is green, and DAPI (DNA) is blue. Phase contrast images are also shown. Arrows show the pericentrosomal localization of CD133. Arrowheads show the predominant nuclear localization of β-catenin. Scale bars: 10 μm. (**B**) Symmetry-breaking of autophagic activity during cytokinesis. During cytokinesis, CD133 endosomes asymmetrically localize to the centrosome and suppress autophagy. In autophagy-repressed daughter cells, the nuclear localization of β-catenin is enhanced and suppresses p62/SQSTM1 gene expression. On the other hand, in daughter cells in which CD133 endosomes do not localize to the centrosome, autophagy is fully activated, and β-catenin remains, localizing to the plasma membrane. In addition, the gene expression of p62/SQSTM1 is promoted. This figure (**B**) was partly modified from the Journal of *Journal of Japanese Biochemical Society* **2022**, *94*, 415–418. doi:10.14952/SEIKAGAKU.2022.940415 [12].

## Data Availability

Not applicable.

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
