# Peer review of "Molecular Regulation of Autophagy and Asymmetric Cell Division by Cancer Stem Cell Marker CD133"

_cells, 2023, doi:10.3390/cells12050819_

Round 1

Reviewer 1 Report

In this commentary manuscript, the authors want to explain the detailed mechanisms of how CD133 and beta-catenin collaborate to regulate asymmetric cell division.

However, there are some points to be clarified:

1. figure 1B seems to have been published/used (almost identical) in "Izumi, J Cancer Biol 2020; 1(1): 7-9, CD133 and centrosomes: How CD133 inhibits autophagy and induces the undifferentiated state of cancer cells at centrosomes" as shown below.

2. In addition, in "Izumi, J Cancer Biol 2020; 1(1): 7-9, CD133 and centrosomes: How CD133 inhibits autophagy and induces the undifferentiated state of cancer cells at centrosomes" the author suggests endosomal-CD133 will be located on centrosome and then inhibits autophagy resulted in undifferentiated of cancer cells. The author didn't discuss the difference between these two concepts, “inducing the undifferentiated state” versus “asymmetric cell division”.

3. in this manuscript, most information was quoted from "Izumi, H.L., Y.;Yasunami, M.;Sato, S.; Mae, T.; Kaneko, Y.; Nakagawara, A. Asymmetric 176 pericentrosomal CD133 endosomes induce the unequal autophagic activity during cytokinesis in 177 CD133-positive human neuroblastoma cells. Stem Cells 2022, in press". Thus, it is difficult to follow the manuscript without getting the above submitted manuscript.

4. Fig 2B the color and illustration are difficult to understand.

5. The cytokinesis is a rapid progression. It also requires a process that nuclear envelope remodeling and the ER to be reshaped. Thus, it needs additional evidence to explain the CD133-beat-catenin collaboration, which might go through transcription regulation to modulate autophagy at that time.

7. For asymmetric cell division, it needs to be involved in the discussion of mother and daughter centrioles/centrosomes.

8. How are naive Huh-7 cells, a hepatocellular carcinoma cell line, and SK-N-DZ cells, a neuroblastoma cell line, as cancer stem cell models?

Author Response

In this commentary manuscript, the authors want to explain the detailed mechanisms of how CD133 and beta-catenin collaborate to regulate asymmetric cell division.
However, there are some points to be clarified:

  1. figure 1B seems to have been published/used (almost identical) in "Izumi, J Cancer Biol 2020; 1(1): 7-9, CD133 and centrosomes: How CD133 inhibits autophagy and induces the undifferentiated state of cancer cells at centrosomes" as shown below.

Thank you for your comments. We rewrote new figure 1B.

  1. In addition, in "Izumi, J Cancer Biol 2020; 1(1): 7-9, CD133 and centrosomes: How CD133 inhibits autophagy and induces the undifferentiated state of cancer cells at centrosomes" the author suggests endosomal-CD133 will be located on centrosome and then inhibits autophagy resulted in undifferentiated of cancer cells. The author didn't discuss the difference between these two concepts, “inducing the undifferentiated state” versus “asymmetric cell division”.

Thank you for your critical suggestion. We added the following sentences in p 7, lane 1.

“Asymmetric cell division is a characteristic of cancer stem cells as well as normal stem cells, and is a very ingenious strategy because it maintains appropriate numbers of self-renewal stem cells and differentiated cells in a single division.”

  1. in this manuscript, most information was quoted from "Izumi, H.L., Y.;Yasunami, M.;Sato, S.; Mae, T.; Kaneko, Y.; Nakagawara, A. Asymmetric pericentrosomal CD133 endosomes induce the unequal autophagic activity during cytokinesis in CD133-positive human neuroblastoma cells. Stem Cells 2022, in press". Thus, it is difficult to follow the manuscript without getting the above submitted manuscript.

Thank you for your comments. Based on your suggestion, we added the following sentences in p 5, lane 6.

“Autophagy is a highly conserved protein/organelle degradation system. It is responsible for the turnover of long-lived proteins and disposal of excess or damaged organelles, in order to maintain cell homeostasis [14] [15]. ULK1 is at the top of the autophagy cascade, and GABARAP (GABA receptor-associated protein) is necessary for the activation of ULK1-mediated autophagy initiation. The activated-ULK1 then activates the autophagy initiation complex, and elongation of the isolation membrane also occurs [15] [16]. The isolation membrane then closes and engulfs cytoplasmic constituents, forming an autophagosome. Finally, the autophagosome fuses with a lysosome, resulting in complete degradation by lysosomal enzymes [14] [17].”

  1. Fig 2B the color and illustration are difficult to understand.

Thank you for your comments. We rewrote new Fig 2B (below).

  1. The cytokinesis is a rapid progression. It also requires a process that nuclear envelope remodeling and the ER to be reshaped. Thus, it needs additional evidence to explain the CD133-beat-catenin collaboration, which might go through transcription regulation to modulate autophagy at that time.

Thank you for your critical comments. There is an important paper (Science 349: 1334 (2015)), which shows the neural stem cells generate a lateral diffusion barrier in the membrane of the endoplasmic reticulum (ER), thereby promoting asymmetric segregation of cellular components.

Unfortunately, the remodeling of nuclear envelope and ER and the transcriptional regulation in our neuroblastoma cells are for further study.

  1. For asymmetric cell division, it needs to be involved in the discussion of mother and daughter centrioles/centrosomes.

Thank you for your suggestion. The centrosome inheritance during asymmetric cell division is very important issue, and our ongoing project. In this time, we cannot describe it. Alternatively, we added the following sentences in p 8, lane 12.

“Alternatively, inheritance of the mother or daughter centrosome might be a key process to determine symmetry-breaking in cell division. The inheritance pattern of mother/daughter centrosomes during asymmetric cell division is an important issue [23] [24], and further study is necessary for its complete elucidation.”

  1. How are naive Huh-7 cells, a hepatocellular carcinoma cell line, and SK-N-DZ cells, a neuroblastoma cell line, as cancer stem cell models?

Yes, SK-N-DZ is a cancer stem-like cells. On the other hand, the study of asymmetric cell division using Huh-7 cells is difficult because there are an only few mitotic cells.

Reviewer 2 Report

Authors report the following article in the bibliography (5) as in press, meanwhile is has been already published by themselves on Stem Cells, Volume 40, Issue 4, April 2022, Pages 371–384, https://doi.org/10.1093 /stmcls/sxac007. In addition they report same images in this commentary whose have been previously reported in the above mentioned article. This reviewer suggest to change the bibliography and to substantially modify images already published on a different journal. In addition, reported results should be analyzed from a different point of view compared to articles that authors already published, otherwise the commentary really is not adding any new perspectives, as as one would expect from a commentary.

Author Response

Authors report the following article in the bibliography (5) as in press, meanwhile is has been already published by themselves on Stem Cells, Volume 40, Issue 4, April 2022, Pages 371–384, https://doi.org/10.1093 /stmcls/sxac007. In addition they report same images in this commentary whose have been previously reported in the above mentioned article. This reviewer suggest to change the bibliography and to substantially modify images already published on a different journal. In addition, reported results should be analyzed from a different point of view compared to articles that authors already published, otherwise the commentary really is not adding any new perspectives, as one would expect from a commentary.

Thank you for your comments.

We refer paper (5) in the following;

  1. Izumi, H.; Li, Y.; Yasunami, M.; Sato, S.; Mae, T.; Kaneko, Y.; Nakagawara, A. Asymmetric Pericentrosomal CD133 Endosomes Induce the Unequal Autophagic Activity During Cytokinesis in CD133-Positive Human Neuroblastoma Cells. Stem Cells 2022, 40, 371-384, doi:10.1093/stmcls/sxac007.

We revised our figure data as following;

In addtition, we added the following sentences in p 8, lane 7 (in conclusion and perspective section) .

“Why do CD133 endosomes become asymmetrically distributed during cell division? Generally, recycling endosomes are present in the midzone in the telophase and are then transported to the centrosome by cytokinesis [22]. The mechanism is not yet known, but it is likely that CD133 endosomes are symmetrically distributed in most cases, but due to some stochastic timing, approximately 20% of cells may show an asymmetric distribution. Alternatively, inheritance of the mother or daughter centrosome might be a key process to determine symmetry-breaking in cell division. The inheritance pattern of mother/daughter centrosomes during asymmetric cell division is an important issue [23] [24], and further study is necessary for its complete elucidation.”

Reviewer 3 Report

This concise article describe the role autophagy machinery in self-renewal of cancer stem cells and molecular marker of CD133. As obvious from reference list and content, it is a concise report and may be considered as a commentary because the topic seems interesting. However, the organization of and knowledge information sufficient and the content provided by the authors should be more updated (only a reverence above 2020 is in the list). Before acceptance, the novel and recent references should be discussed and the content should be improved.

 My suggestion: texts and figure related to autophagy mechanism and its machinery is missing and should be firstly included in the paper. The authors can briefly describe molecular connections between autophagy and cancer stem cells, then connect it with CD133. This help readers to better understand a connections between CD133 and autophagy regulatory proteins.

Author Response

This concise article describes the role autophagy machinery in self-renewal of cancer stem cells and molecular marker of CD133. As obvious from reference list and content, it is a concise report and may be considered as a commentary because the topic seems interesting. However, the organization of and knowledge information sufficient and the content provided by the authors should be more updated (only a reverence above 2020 is in the list). Before acceptance, the novel and recent references should be discussed and the content should be improved.

My suggestion: texts and figure related to autophagy mechanism and its machinery is missing and should be firstly included in the paper. The authors can briefly describe molecular connections between autophagy and cancer stem cells, then connect it with CD133. This help readers to better understand a connection between CD133 and autophagy regulatory proteins.

Thank you for your suggestion.

At first, we added the following sentences in introduction section (p 5, lane 5).

“Autophagy is a highly conserved protein/organelle degradation system. It is responsible for the turnover of long-lived proteins and disposal of excess or damaged organelles, in order to maintain cell homeostasis [14] [15]. ULK1 is at the top of the autophagy cascade, and GABARAP (GABA receptor-associated protein) is necessary for the activation of ULK1-mediated autophagy initiation. The activated-ULK1 then activates the autophagy initiation complex, and elongation of the isolation membrane also occurs [15] [16]. The isolation membrane then closes and engulfs cytoplasmic constituents, forming an autophagosome. Finally, the autophagosome fuses with a lysosome, resulting in complete degradation by lysosomal enzymes [14] [17].”

 (We subsequently investigated the biochemical functions of centrosome-localized CD133, and found that it traps GABARAP, an important regulator of autophagy initiation, to inhibit GABARAP-mediated ULK1 activation, and the subsequent initiation of autophagy (Figure 1B) [4].)

Reviewer 4 Report

The commentary entitled “Molecular regulation of autophagy and asymmetric cell division by cancer stem cell marker CD133” by Hideki Izumi et al. gives a short commentary of the recent study published on Stem Cells journal and entitled "Asymmetric Pericentrosomal CD133 Endosomes Induce the Unequal Autophagic Activity During Cytokinesis in CD133-Positive Human Neuroblastoma Cells" by same authors.

The manuscript is well written and discuss about the potential role of CD133 in inducing the symmetry breaking of autophagic activity during cyto-kinesis and that autophagy based-asymmetric cell division is one of the driving forces of tumor cell heterogeneity. Thus, pericentrosomal CD133 seems to have the unique role of regulating autophagy and the authors suggests that autophagy-based asymmetric cell division plays an important role in tumor cell heterogeneity.

 Specific Comments Added:

As already reported, in my opinion, this commentary is suitable for publication in the present form.
Accordingly, since this manuscript is a commentary, it is a  comment on the  recent newly published article by Izumi, H.L., Y.;Yasunami, M.;Sato, S.; Mae, T.; Kaneko, Y.; Nakagawara, A. entitled "Asymmetric 176 pericentrosomal CD133 endosomes induce the unequal autophagic activity during cytokinesis in 177 CD133-positive human neuroblastoma cells. " Stem Cells, 2022, 40, 371–384 https://doi.org/10.1093/stmcls/sxac007. Advance access publication 14 March 2022.
For these reasons  I think that it is not possible to suggest specific improvements regarding the methodology and further controls. Moreover the conclusion appeared to be consistent with the previous findings already published, as well as the references, the figures and the conclusions which are consistent with the evidence and arguments presented  addressing the main question posed.

Author Response

The commentary entitled “Molecular regulation of autophagy and asymmetric cell division by cancer stem cell marker CD133” by Hideki Izumi et al. gives a short commentary of the recent study published on Stem Cells journal and entitled "Asymmetric Pericentrosomal CD133 Endosomes Induce the Unequal Autophagic Activity During Cytokinesis in CD133-Positive Human Neuroblastoma Cells" by same authors.

The manuscript is well written and discuss about the potential role of CD133 in inducing the symmetry breaking of autophagic activity during cyto-kinesis and that autophagy based-asymmetric cell division is one of the driving forces of tumor cell heterogeneity. Thus, pericentrosomal CD133 seems to have the unique role of regulating autophagy and the authors suggests that autophagy-based asymmetric cell division plays an important role in tumor cell heterogeneity.

Specific Comments Added:

As already reported, in my opinion, this commentary is suitable for publication in the present form.
Accordingly, since this manuscript is a commentary, it is a comment on the recent newly published article by Izumi, H.L., Y.;Yasunami, M.;Sato, S.; Mae, T.; Kaneko, Y.; Nakagawara, A. entitled "Asymmetric 176 pericentrosomal CD133 endosomes induce the unequal autophagic activity during cytokinesis in 177 CD133-positive human neuroblastoma cells. " Stem Cells, 2022, 40, 371–384 https://doi.org/10.1093/stmcls/sxac007. Advance access publication 14 March 2022.

For these reasons I think that it is not possible to suggest specific improvements regarding the methodology and further controls. Moreover the conclusion appeared to be consistent with the previous findings already published, as well as the references, the figures and the conclusions which are consistent with the evidence and arguments presented addressing the main question posed.

Thank you very much for your comments because you understand our study very well.

Round 2

Reviewer 1 Report

The manuscript has significantly improved. However, the authors need to address the following issues:

1. This reviewer does not agree with the response to question #5 as shown below. The experimental data are important to support the finding of this manuscript.

  1. The cytokinesis is a rapid progression. It also requires a process that nuclear envelope remodeling and the ER to be reshaped. Thus, it needs additional evidence to explain the CD133-beat-catenin collaboration, which might go through transcription regulation to modulate autophagy at that time.

Thank you for your critical comments. There is an important paper (Science 349: 1334 (2015)), which shows the neural stem cells generate a lateral diffusion barrier in the membrane of the endoplasmic reticulum (ER), thereby promoting asymmetric segregation of cellular components.

Unfortunately, the remodeling of nuclear envelope and ER and the transcriptional regulation in our neuroblastoma cells are for further study.

2. Though the study is interesting and novel, there are deficiencies in demonstrating that transcription regulation of beta-catenin is the key step for controlling autophagy to modulate asymmetric cell division. The conclusion, as shown in figure 2B, is preliminary. Additional evidence is required.

  minor questions: 1. What are the differences between the upper panel and the middle panel of the control siRNA in Figure 2A? Why it showed different CD133 and beta-catenin locations in these two panels? The location of CD133 is endosome or centrosome? Non-nuclear located beta-catenin is on the membrane or in the cytoplasm.    2. As the response from the authors,  "8. How are naive Huh-7 cells, a hepatocellular carcinoma cell line, and SK-N-DZ cells, a neuroblastoma cell line, as cancer stem cell models? Yes, SK-N-DZ is a cancer stem-like cells. On the other hand, the study of asymmetric cell division using Huh-7 cells is difficult because there are an only few mitotic cells." What does it mean "Huh-7 cells are only a few mitotic cells"? And if naive Huh7 cells are not cancer stem(-like) cells, and SK-N-DZ is CD44 high expressed cell which may exist a cancer stem-like phenotype. The description in lines 73-74 will be questionable "As a result, we identified Huh-7, a hepatocellular carcinoma cell line, and SK-N-DZ, a neuroblastoma cell line, as cancer stem cell models."  

Author Response

The manuscript has significantly improved. However, the authors need to address the following issues:

  1. This reviewer does not agree with the response to question #5 as shown below. The experimental data are important to support the finding of this manuscript.

Unfortunately, since this is a commentary on a paper published in Stem Cells, I did not have time to examine additional experiments. Your request goes beyond a commentary paper. Although we did not address ER and nuclear envelope remodeling, we have data that midbody remnant is preferentially inherited to CD133-positive daughter cells with low autophagy during cytokinesis (unpublished data). The inheritance of midbody remnant during cytokinesis is as important a topic as nuclear membrane and ER remodeling. Anyway, I hope to analyze them in the near future with your suggestion.

  1. Though the study is interesting and novel, there are deficiencies in demonstrating that transcription regulation of beta-catenin is the key step for controlling autophagy to modulate asymmetric cell division. The conclusion, as shown in figure 2B, is preliminary. Additional evidence is required.

Thank you for your comment. As a next step, I would like to deepen my research using TOPFlash/ FOPFlash vectors (beta-catenin reporters) with your opinion.

minor questions:

  1. What are the differences between the upper panel and the middle panel of the control siRNA in Figure 2A? Why it showed different CD133 and beta-catenin locations in these two panels? The location of CD133 is endosome or centrosome? Non-nuclear located beta-catenin is on the membrane or in the cytoplasm.   

The upper panels in Figure 2A show symmetric cell division, and the middle panels in Figure 2A show asymmetric cell division. The location of CD133 is pericentrosomal endosome. Non-nuclear located beta-catenin is on the membrane because the staining pattern of non-nuclear located beta-catenin consists with that of Na/K-ATPase (a marker of plasma membrane) (figure 7B in Stem Cells 40: 371 (2022)).

  1. As the response from the authors,  "8. How are naive Huh-7 cells, a hepatocellular carcinoma cell line, and SK-N-DZ cells, a neuroblastoma cell line, as cancer stem cell models?Yes, SK-N-DZ is a cancer stem-like cells. On the other hand, the study of asymmetric cell division using Huh-7 cells is difficult because there are an only few mitotic cells." What does it mean "Huh-7 cells are only a few mitotic cells"? And if naive Huh7 cells are not cancer stem(-like) cells, and SK-N-DZ is CD44 high expressed cell which may exist a cancer stem-like phenotype. The description in lines 73-74 will be questionable "As a result, we identified Huh-7, a hepatocellular carcinoma cell line, and SK-N-DZ, a neuroblastoma cell line, as cancer stem cell models."

Thank you for your comments. Huh-7 cells also show asymmetric division, but there are fewer cells in mitosis, making analysis difficult. In addition, we have not examined the expression of CD44, another cancer stem cell marker.

Reviewer 2 Report

Authors almost fulfilled requests from this reviewer. However fig 2B is still very similar to images already published on Stem Cells. 

Author Response

Authors almost fulfilled requests from this reviewer. However fig 2B is still very similar to images already published on Stem Cells. 

Thank you for your opinion. According to your suggestion, I have modified fig. 2B.

Reviewer 3 Report

Accepted in the current form!

Author Response

Accepted in the current form!

Thank you for your review.